# Thermal and Structural Analysis of Epoxidized Jatropha Oil and Alkaline Treated Kenaf Fiber Reinforced Poly(Lactic Acid) Biocomposites

**DOI:** 10.3390/polym12112604

**Published:** 2020-11-06

**Authors:** Siti Hasnah Kamarudin, Luqman Chuah Abdullah, Min Min Aung, Chantara Thevy Ratnam

**Affiliations:** 1School of Industrial Technology, Faculty of Applied Sciences, Universiti Teknologi MARA (UiTM), Shah Alam 40450, Selangor, Malaysia; 2Higher Education Centre of Excellence (HICoE), Institute of Tropical Forestry and Forest Products (INTROP), Universiti Putra Malaysia (UPM), UPM Serdang 43400, Selangor, Malaysia; minmin_aung@upm.edu.my; 3Department of Chemical Engineering, Faculty of Engineering, Universiti Putra Malaysia (UPM), UPM Serdang 43400, Selangor, Malaysia; 4Unit Chemistry, Centre of Foundation Studies for Agricultural Science, Universiti Putra Malaysia (UPM), UPM Serdang 43400, Selangor, Malaysia; 5Radiation Processing Technology Division, Malaysian Nuclear Agency, Bangi 43000, Selangor, Malaysia; chantara@nuclearmalaysia.gov.my

**Keywords:** kenaf, poly(lactic acid), epoxidized jatropha oil, DSC, TGA, SEM, FTIR, alkaline treatment

## Abstract

New environmentally friendly plasticized poly(lactic acid) (PLA) kenaf biocomposites were obtained through a melt blending process from a combination of epoxidized jatropha oil, a type of nonedible vegetable oil material, and renewable plasticizer. The main objective of this study is to investigate the effect of the incorporation of epoxidized jatropha oil (EJO) as a plasticizer and alkaline treatment of kenaf fiber on the thermal properties of PLA/Kenaf/EJO biocomposites. Kenaf fiber was treated with 6% sodium hydroxide (NaOH) solution for 4 h. The thermal properties of the biocomposites were analyzed using a differential scanning calorimetry (DSC) and thermogravimetric analysis (TGA). It must be highlighted that the addition of EJO resulted in a decrease of glass transition temperature which aided PLA chain mobility in the blend as predicted. TGA demonstrated that the presence of treated kenaf fiber together with EJO in the blends reduced the rate of decomposition of PLA and enhanced the thermal stability of the blend. The treatment showed a rougher surface fiber in scanning electron microscopy (SEM) micrographs and had a greater mechanical locking with matrix, and this was further supported with Fourier-transform infrared spectroscopy (FTIR) analysis. Overall, the increasing content of EJO as a plasticizer has improved the thermal properties of PLA/Kenaf/EJO biocomposites.

## 1. Introduction

The demand for plastics as one of the most highly valued materials has benefited tremendously due to their extraordinary versatility, manufacturability and low cost in price [1]. However, most of the plastics available are coming from synthetic polymers where the product comes at the price of depleting fossil fuel and adverse effects on the environment. Plastics waste disposal problems create environmental pollution. Therefore, the high rate of depletion of petroleum resources, growing ecological, social and economic awareness and new environmental regulations have stimulated the attention of many researchers to search for the potential usage of green materials to replace plastics, such as bioplastics that are environmentally friendly and available resourcefully to meet the ever increasing demand for plastics [2,3,4]. Among all bioplastics available, poly(lactic acid) (PLA) is the most promising biodegradable polymer to be developed due to its versatility and immense potential to be used in various industrial applications [5,6]. The utilization of biodegradable polymers such as PLA in the market is progressively increasing due to its relevant physico-chemical properties, together with its biosourced origin, biodegradability and even biocompatibility [7]. Nevertheless, unmodified PLA are not directly suitable for durable applications due to their low heat resistance, low impact resistance, low crystallinity/crystallization kinetics, brittleness and stiffness. Hence, the addition of plasticizer is needed in order to improve the flexibility, elongation and impact properties of composites [8,9]. Thus, it is important to look for opportunities and provide a sustainable plasticizer for PLA due to the realization of the biodegradability and renewability properties of polymers. Epoxidized vegetable oils have received much attention because they act as reactive plasticizers through the reactivity between epoxy functional groups and the –OH and –COOH groups of PLA polymers [10,11,12,13]. Epoxidized vegetable oils can be produced through the epoxidation of C=C bonds of oils and plasticizers, without volatility and also allow low migration tendency [12,14,15]. Modified vegetable oils such as corn, olive, canola, cottonseed, linseed, palm, rapeseed and soybean oils are of renewed interest as substitutes for phthalates and have been applied as plasticizers or stabilizers for polymers by means of structural and thermal studies [12,16,17,18]. These oils are categorized as edible oils and could cause negative impacts to the world such as the depletion of food supply, leading to economic imbalance.

Manufacturing high performance composites from natural fiber resources and nonedible vegetable oil is relatively easy and possesses many environmental benefits as an ambitious goal currently being pursued by researchers across the globe—these composites are usually termed as “green composites” [19,20,21]. Most of the natural fibers in composites are lighter, due to their density with other synthetic fibers and materials, inexpensive, a readily available source of lignocellulosic biomass, produced in billions of tons around the world, and abundant in nature to support the viability of a circular economy via reducing the carbon footprint as well as increasing the mechanical performance of the biopolymers. Natural fibers can be readily purchased as cheap as $0.50/kg as compared to expensive glass fiber ($3.25/kg), plus are easily grown in just a few months [22]. Natural fibers serve as a very good source for cellulose as they grow easily and are also gaining importance nowadays as a cash crop for local farmers. Natural fibers are also significantly lighter than glass, with a density of 1.15 to 1.50 g/cm^3^ versus 2.4 g/cm^3^ for e-glass [23].

As a result of its durability, reliability, sustainability, biocompatibility and lightness together with excellent mechanical properties, natural fibers are likely to replace and reduce the dependency on synthetic fibers, and become the next generation of value-added ecotechnology for various applications, primarily in building, automotive parts and construction etc. [24,25,26]. Research on natural fiber-reinforced composites in various different types of polymers has been carried out by many researchers which include bamboo [27,28], cotton [29,30], flax [31,32], jute [33,34], ramie [35,36], sisal [37,38], pineapple leaf [39], hemp [40,41], kenaf [42,43] and wheat straw pulp [44]. The high value of cellulose content in kenaf has been recognized for providing strength and stiffness in their utilization in several industrial applications and nanomaterial products [45,46].

Surface modification and treatment of the fibers are necessary to reduce the hydrophilicity and minimize the interfacial energy with hydrophobic polymers. In case of surface treatment, chemical treatment such as alkali treatment, bleaching, acetylation, benzoylation, vinyl grafting, peroxide treatment and treatment with various coupling agents leads to an increase in the amount of interfacial adhesion between the natural fiber and the matrix, thus enhancing the mechanical properties of the fiber [47]. Among the chemical treatments listed, one method of effective surface modification is alkali treatment which has been used by several researchers to improve the performance of composites [3,48,49].

The alkali treatment or mercerization introduces new moieties that can effectively penetrate the matrix quickly and remove a certain amount of hemicellulose, lignin, wax and oils that cover the external surface of the fiber cell wall. As a result of NaOH penetration, this modification alters the fiber and turns it into rougher surfaces for greater fiber interlocking and matrix penetration, as well as a larger contact area between the fiber and the matrix [50]. Alkali treatment removes the noncellulosic components in fiber such as hemicellulose and lignin, consequently producing a close-packed cellulose compound. The close-packed cellulose is linked to each other through hydrogen bonding and alkaline treatment; thus, the crystallinity of the fiber increases after undergoing the treatment process [51]. In other words, the ester group in polymers could react with OH groups after NaOH treatment because the fibers have more available reactive sites for the ester group. The crosslinking reconstructed the fibers and made them stronger and able to transfer more load to the matrix.

Biocomposites consist of two or more distinct constituents or phases when combined together resulting in a material with entirely different properties than those of the individual components. Several studies have revealed that through the fabrication biocomposites of natural fiber and poly(lactic acid) has improved the thermal properties of the composites. Among the biocomposites that combine natural fibers with poly(lactic acid) are sugar palm [52], kenaf [53], sisal [54], banana [55], jute [56], basalt [57], flax [58], coir [59], oil palm fiber empty fruit bunch (EFB) [60] and pineapple leaf fiber [61].

Research on the development of polylactic acid reinforced kenaf fiber biocomposites has indicated that the value of storage modulus tends to be higher with the addition of kenaf fiber. Tan delta and mechanical loss factor decreased with kenaf fiber content, which can be associated with the compatibility of the PLA matrix and kenaf fiber [62]. However, there is a lack of literature on alkaline treated natural fiber-reinforced epoxidized jatropha oil (EJO)/PLA biocomposites. Previous articles from our laboratory have reported on the mechanical and physical properties of kenaf-reinforced poly(lactic acid) plasticized with epoxidized jatropha oil [3]. In this study, the effect of epoxidized jatropha oil and alkaline-treated fiber on the thermal properties of PLA biocomposites will be investigated. PLA/Kenaf/EJO biocomposites have been successfully developed as a new environmentally friendly super material with improved properties that are also cost effective to replace the glass fiber composites. This type of biocomposite complies with the requirements of after-use management of the composites that would not be harmful to the environment. The findings from this study will pave the way towards a greater usage of vegetable oil through epoxidation as well as natural fiber via melt blending for the commercialization of biocomposites.

## 2. Experimental

### 2.1. Materials

In this research, the main material used of poly(lactic acid) (PLA) polymer resins in pellet form, grade 2003D, was obtained from Nature Works, LLC (Minnetonka, MN, USA), and their properties are tabulated in Table 1. Meanwhile, crude jatropha oil was supplied by Bionas Sdn Bhd (Kuala Lumpur, Malaysia) as a nonfood grade material and was used as received. Kenaf fiber, grade V36 (a variety of kenaf species that planted in Malaysia) in the size of 40 µm mesh was obtained from the National Kenaf and Tobacco Board (Kelantan, Malaysia).

### 2.2. Epoxidation of Jatropha Oil

The epoxidation of jatropha oil was carried out by the in-situ epoxidation method described by Saalah et al. [63] with a slight modification in the molar ratio of 1:0.6:1.7 (organic acid: double bonds: hydrogen peroxide) at a temperature of 60 °C for 5 h reaction time. A preweighed amount of formic acid was slowly added into a 1 L 4-neck flask where the desired jatropha oil had been prepared and the solution mixture was then heated to 40 °C under continuous stirring in a water bath. Hydrogen peroxide 30% was then added dropwise to the solution before the reaction temperature was raised up to 60 °C. The epoxidation process was carried out in a closed fume hood.

### 2.3. Alkaline Treatment of Kenaf Fiber

The mercerization of kenaf fiber was treated with 6% NaOH solution to modify the fiber’s surfaces. The kenaf fibers were alkali treated and soaked in hot distilled water with continuous stirring at 60 °C for 4 h to remove wax and other impurities. Then the treated fibers were washed several times with distilled water until pH 7 was obtained. Subsequently, the fibers were dried in an oven at 60 °C for 24 h.

### 2.4. Preparation of Biocomposites

For the fabrication of biocomposites, the ratio between the PLA matrix was from 65 to 69 wt.% and the kenaf fiber reinforcement was selected at 30 wt.% (Table 2). Prior to mixing, PLA was dried for 2 h at 80 °C to prevent hydrolytic degradation during processing. PLA was premixed with 1–5 wt.% of epoxidized jatropha oil and 30 wt.% of kenaf fiber. A total of 30 wt.% of kenaf fiber was chosen as a suitable amount of fiber loading for the reinforcement of PLA and EJO due to the strong mechanical interlocking between the kenaf fiber and the matrix. The fibers were well distributed and had a great interaction with the matrix. The amount of matrix was sufficient enough to wet out the fibers and fully transfer the stress effectively at such fiber loadings. PLA/Kenaf/EJO biocomposites were prepared by the melt blending technique using a Brabender internal mixer (Dushburg, Germany) at 170 °C for 10 min at 50 rpm motor speed. Then, the blended materials were subjected to hot and cold press for compression molding with a pressure of 10.8 MPa at 170 °C for 11 min to produce biocomposite sheets of uniform 1 mm and 3 mm thickness. The biocomposites were then analyzed for characterization of their thermal properties.

### 2.5. Kenaf Fiber Analysis

The structural changes of the fiber before and after the alkaline treatment was determined using Fourier transform infrared spectroscopy (FTIR) analysis performed using a Perkin Elmer: Model 1000 series, Akron, OH, USA. The device had an attenuated total reflectance (ATR) attachment with a diamond crystal. Measurements were made with 1 cm^−1^ resolution in the wavenumber range from 4000 to 400 cm^−1^.

The surface morphology of the fiber was investigated by using a JEOL model JSM-6300F, North Billerica, MA, USA scanning electron microscope (SEM).

The chemical compositions of the untreated and treated kenaf fiber were analyzed according to the following standard methods: ethyl benzene extractive (TAPPI T 204 CM-97), lignin (TAPPI T 222 OM-98), holocellulose [64] and alfa-cellulose (TAPPI T 203 CM-99).

### 2.6. Biocomposites Analysis

A TG Analyzer (Perkin Elmer TGA7, Akron, OH, USA) was used for the evaluation of the thermal behaviors and stability of fiber, PLA and the biocomposites. Thermogravimetric analysis (TGA) was performed via a TGA Q500 from TA Instruments, New Castle, DE, USA and was conducted under ramp mode from 30 to 600 °C under a nitrogen atmosphere at a flow rate of 50 mL/min. The heating rate utilized was 20 °C/min. A sample of 5–10 mg of the materials was heated in the sample pan. The temperature was raised. The determination of the percentage of weight loss versus temperature was analyzed from the TGA curve.

A differential scanning calorimetry (DSC) test was performed using a DSC (Mettler-Toledo, Model: DSC 823e, Greifensee, Switzerland). Approximately 5.0 mg of sample was weighed in an aluminum pan and subjected to 2 heating-cooling cycles. The degree of crystallinity of composite (*X*_c_) was determined according to Equation (1):*Xc* = ((∆*Hm*−∆*H*_cc_)/(∆*H*100% × WPLA)) × 100(1)
where ∆*H*100% is the melt enthalpy for theoretical 100% crystalline PLA, ∆*H*_cc_ is the enthalpy of cold crystallization process, ∆*H*_m_ is the enthalpy of melting process, and WPLA is the weight fraction of PLA. In this work, a value of 93 J/g was taken as the melt enthalpy of 100% crystalline PLA [65,66,67]. The temperature was set from room temperature up to 500 °C and the heating rate for the sample was about 10 °C/min. The melting point (*T*_m_) and the glass transition temperature (*T*_g_) of the materials was determined.

## 3. Results and Discussion

### 3.1. Fourier Transform Infrared (FTIR) Spectroscopy Analysis

Figure 1 shows the FTIR spectra for both untreated and treated kenaf fibers in the fingerprint region between 4000 to 400 cm^−1^ range. The observed spectra can be classified into few regions which correspond to the following peaks band assignment: –OH stretching (3325 cm^−1^), –CH–vibration (2897 cm^−1^), –C=O stretching (1732 cm^−1^), –C=C– stretching (1641 cm^−1^), –CH–bending (1369 and 1235 cm^−1^), –C–C– stretching (1156–1031 cm^−1^), –CH–stretching (897 cm^−1^) and –OH (600 cm^−1^). It was observed that both untreated kenaf and NaOH treated kenaf showed a broad peak of alcoholic O–H stretching absorptions at approximately 3300 cm^−1^. Nonetheless, treated fiber had a lower peak intensity which may be attributed to the removal of hemicellulose and lignin. Scientifically, natural fibers are rich with hydroxyl groups that are available on the fiber surface of kenaf fibers. Alkaline treatment breaks the hydrogen bonding between the hydroxyl groups (–OH) of the cellulose, hemicellulose and lignin that reacted with sodium hydroxide. Thus, this phenomenon will lead to the defibrillation, the breakdown of fiber bundles into smaller fibers. This was observed by the reduction and broadening of the peaks around 3200 cm^−1^. The absorbance peaks around 2900–2800 cm^−1^ were associated with the stretching of the C–H group. More specifically, it was found that the vibration peak of C–H stretched at 2900 cm^−1^ in cellulose and hemicellulose present in the untreated fiber became weak, attributed to the part of the hemicellulose being removed [68].

The absorption peak was observed at approximately 1729.8 cm^−1^, in the untreated kenaf spectra, however this peak was not present in the spectrum for treated fiber. According to Tserki et al. [69] and Sgriccia et al. [70], this band is attributed to the C=O stretching of the acetyl group in hemicellulose. Moreover, [71] had assigned the peak as the ester linkage of the carboxylic group in the ferulic and p-coumaric acid of lignin and/or hemicellulose. The disappearance of this peak for treated fiber indicated the elimination of hemicellulose and lignin as a result of the mercerization process. Mercerization removes the waxy layer, adhesive pectins and hemicelluloses that bind fiber bundles to each other and to the pectin and hemicellulose-rich sheets of the core [72]. Furthermore, the intensity of the peak at 1506.13 cm^−1^ for the untreated fiber decreased after the alkalization treatment of fiber took place. The intensity of the peak which was attributed to the hydroxyl group was reduced in the case of the treated kenaf fiber. The peak was corresponding with the C=C aromatic symmetrical stretching for lignin [73]. The spectroscopy peaks in range of 1380–1320 cm^−1^ for both untreated and treated fiber exhibited the bending vibration of C–H and C–O groups of the aromatic ring in polysaccharides. There was a significant change in a peak at absorbance 1230.3 cm^−1^ for untreated fiber which was found to be shifted to 1319 cm^−1^ after the treatment process and a decrement in the intensity of the peak was also observed. The shifted peak was associated with the removal of lignin after treatment of the kenaf fiber, which was attributed to the aryl group in lignin, and thus the change had been detected [74]. Moreover, after chemical treatment, the intense peaks appeared around the 1070–1010 cm^−1^ region which was attributed to the stretching of C–O and O–H groups [75]. These results had confirmed that the chemical treatment of kenaf fiber had been successfully achieved. Table 3 shows the wavenumbers representing bond type for other types of natural fibers for comparison.

### 3.2. Scanning Electron Microscopy (SEM) Analysis

SEM micrographs of untreated kenaf fibers (UTK) at different magnifications are shown in Figure 2a–c. It can be seen that a mixture of different size of UTK with some of the fibers were being twisted and damaged as presented in Figure 2a. Detached fibers in the UTK sample could be due to the mechanical procedure undertaken during fiber separation and extraction processes. Fibers with different lengths and diameters are also evident from Figure 2a. A magnified view of UTK showed an uneven surface with slight surface debris (Figure 2b,c). It was observed that the smooth surfaces can be attributed to the fact that the amount of impurities were still on untreated fibers surface, which provided poor interlocking to the polymer matrix, thus, decreasing the mechanical properties [76,77]. Figure 3 shows the cross section of untreated kenaf fibers (UTK). It was obvious that UTK consisted of a bundle of fibers and the diameter of the single fibers was nonuniform. Naturally, these single fibers are held together by a lignin and hemicellulose matrix [78,79]. It is also notable that a single kenaf fiber has a lumen. Lumens are capillary type structure which functions as water, liquid and nutrient transport along the fiber [80]. [80] suggested that filling the lumens during compounding with a polymer matrix is one of the keys in optimizing composite properties. As for the treated kenaf fiber (TK), there were some physical changes in the fiber surface morphology that could be observed as the fiber underwent the alkali treatment process. Relatively, the micrograph image of the treated fiber revealed an improvement in the fiber surface morphology after 6% NaOH treatment. Certain amounts of material such as hemicellulose, lignin, wax, oils and other impurities from the surface of the fiber had successfully been removed with NaOH treatment and can be observed in Figure 4. When the fibers were treated with NaOH, it removed the undesirable materials, thus more reactive sites such as hydroxyl groups of cellulose were revealed [81]. As a result, the treated surface of fiber became rougher with more exposure of fibril as compared to the untreated fiber. In addition, the fibers were then split into finer fibers which in turn led to the higher interlocking and greater adhesion between the fibers and matrix. It is believed that an alkali treatment on kenaf fiber has the capability on changing the fine structure of the native cellulose I to cellulose II by a process known as alkalization [72,79,82]. The reaction of NaOH with kenaf fiber is presented in the following equation:Fiber–OH + NaOH → Fiber–O^−^Na^+^ _H_2_O (surface impurities)(2)

Consistent with other findings of this research, it is proposed that after NaOH treatment, the undesirable surface impurities were removed, thus making them become more rigid. As a result, more rigid fibers had a better ability to transfer the load to the matrix. In other words, the ester group in PLA could react more with the OH groups after NaOH treatment because the fibers had more available reactive sites for the ester groups. From this action, the crosslinking reconstructed the fibers and made them stronger and able to transfer more load to the matrix. As a consequence, it affected the properties of the fibers and fiber-matrix interlocking thus giving a positive effect on fiber-matrix interfacial bonding of composites [81].

### 3.3. Chemical Composition Analysis

The main components of natural fibers are composed of cellulose, hemicellulose, lignin, pectin and waxes. The chemical composition of kenaf fibers had undergone modifications after being chemically treated with an alkali solution. The comparison of the chemical compositions between untreated and treated kenaf fibers is tabulated and presented in Table 4. As can be seen from Table 4, NaOH treatment was found to be effective in removing the lignin and hemicellulose from the kenaf fibers; as the lignin content decreased from 17 to 14% in treated kenaf fiber while the amount of hemicellulose was reduced from 20 to 14%. Meanwhile, the results obtained from the chemical analysis after the treatment showed that the cellulose content increased from 51 to 56%. It is understood that the cellulose content had a greater influence on the properties of kenaf fibers. In other words, the higher the cellulose content in fiber, the greater the stress transfer between fibers will occur, which promotes the higher mechanical properties of the composites in the end. Compared to our earlier studies, the impact properties of biocomposites in this study had been significantly improved after alkali treatment of fibers with over 10 times more than the neat polymer [3]. Moreover, Yahaya et al. [83] had reported that the alkaline treatment modification had promoted the fiber surface adhesion by removing natural and artificial impurities, which led to an enhancement of wettability between fiber and matrix. Chemical modification of natural fibers had disrupted the hydrogen bonding in the network structure, in which it led to the increasing surface roughness of the fiber. This contributed to the improvement in the surface morphology properties of the treated kenaf fiber as compared to the untreated kenaf fiber, which can be seen in Figure 4. Certain amounts of hemicelluloses, lignin, wax and oils covered the external surface of the fiber cell wall; depolymerized cellulose had been removed and exposed the short length crystallites [84]. Additionally, alkaline treatment led to the removal of waxy layers and fibrillation of the fiber bundles into small fibers. Thus, the fiber diameter was reduced, which can be seen in Figure 5b, hence, increasing the aspect ratio. Godavarti et al. [85] demonstrated that they reported a diameter change in their work after the alkaline treatment process. The fiber weight decreased up to 14.6% as a result of the alkaline treatment. This reduction in the fiber diameter was due to the removal of waxy impurities, hemicellulose and lignin after the alkaline treatment. Moreover, the effect of the alkaline treatment or mercerization on the fiber diameter had also been studied by Mbada et al. [84] where the diameters of kenaf fibers had decreased gradually when the fibers had been treated with 1–6 wt.% of NaOH concentration. The smallest fiber diameter in their study was observed for 6 wt.% NaOH concentration, in which it represents the same optimum NaOH concentration used in this study for the treatment of the kenaf fiber.

### 3.4. Thermogravimetric Analysis Properties

TGA is used to measure the mass change, thermal decomposition and thermal stability of the materials in the temperature range over which the materials can be used to the point of noticeable degradation [86,87,88,89,90]. It has been reported by Saba et al. [87] that a major degradation step is seen for neat poly(lactic acid) composites by statistical chain rupture, in which styrene is the primary product in the range of 360–400 °C. The peak appears at around 350–400 °C for neat poly(lactic acid) composites showing the temperature of the maximum degradation is in line with the findings of other researchers [87]. The improvement in the thermal stability of the PLA/Kenaf/EJO blends due to the alkalization on its polymeric blends can be clearly seen by measuring their thermal decomposition temperature at the onset and end of weight loss. The difference in the decomposition temperature between treated and untreated kenaf fiber and their polymer blend composites could be clearly observed. A higher decomposition temperature means that there is an improvement in the thermal stability for the blends.

The thermal stability of all samples of PLA and PLA/TK/EJO biocomposite samples had been analyzed using TGA. The TGA test was conducted in N2 atmosphere with temperature ranging between 25 °C to 600 °C. The weight loss (TG) and its derivative (DTG) curves of PLA and all PLA/TK/EJO biocomposite samples for various EJO loading had been shown in Figure 6 and Figure 7. The onset degradation temperature (*T*_onset_) of PLA was recorded at 315 °C and the degradation completed at 377.83 °C. This was in agreement with similar findings reported by other researchers. Wu et al. [88] reported in their study on PLA/PCL blends that the onset degradation temperature of PLA was recorded at 280 °C and the degradation temperature was recorded at 350 °C.

As can be seen in Figure 6, all PLA/TK/EJO samples had only one stage degradation process above 250 °C. The mass loss step at 190 °C to 330 °C corresponding to the degradation of lignin was not seen in TGA curves of all PLA/TK/EJO biocomposite samples. A further increment in degradation temperature of all PLA/TK/EJO biocomposite samples was observed with the increasing in EJO plasticizer content loading. However, the thermal stability of all PLA/TK/EJO biocomposite samples were lower as compared to PLA.

This is most likely attributed to the fact that the treated kenaf fiber has a lower decomposition temperature than PLA that possibly enhanced the degradation of the crystalline structure of PLA, thus, resulted in the reduction of thermal stability of the composites. The addition of natural fiber has decreased the thermal stability of PLA, since some portion of PLA is replaced with less thermal stable natural fiber, such as kenaf fiber. Furthermore, it might be due to the decrease of a relative molecular mass of PLA. This finding is in agreement with Kalam et al. [91] who observed a similar trend in TGA for PP-filled OPEFB fiber composites. For a clearer observation, the results for TGA analysis for PLA and PLA/TK/EJO biocomposite samples is summarized and tabulated in Table 5.

The thermal stability profile can be expressed as well in terms of parameters such as the initial and final decomposition temperature as well as rapid decomposition temperature. Figure 7 demonstrates a thermal stability profile for each sample via derivative percentage weight thermograms (DTG). From the figure, the pure PLA matrix decomposed most rapidly at 357.85 °C. Upon blending of PLA, EJO and TK, the recorded *T*_max_ was noted with a lower value for all of PLA/TK/EJO biocomposites, where sample of 1 wt.% of EJO content recorded 322.15 °C value. *T*_max_ further increased to 336.94 °C with the addition of 5 wt.% of EJO, which further showed that there was an improvement in thermal stability for the biocomposites. The thermal analysis of the PLA/TK/EJO samples is important as the treatment completed onto kenaf fiber could affect their thermal stability which may influence the degradation process during the composite’s fabrication [92]. Similar studies were reported by Sreekala et al. [93]. Sreekala et al. [93] in which they presented an improvement in thermal stability from TGA analyses for both oil palm empty fruit bunch and oil palm mesocarp fiber after alkali, silane and acetylation treatment, respectively. Moreover, it can be observed in Figure 7 that the increase of EJO content had significantly increased the thermal decomposition rate of PLA/TK/EJO biocomposites at a constant temperature. PLA/TK/EJO 5 wt.% biocomposite shows the highest thermal stability among all biocomposites (1 to 5 wt.% EJO) with the highest initial and final decomposition temperature at 295.99 °C and 382.17 °C, respectively. It also showed the highest *T*_max_ at 336.94 °C. The thermal stability of PLA decreased with the addition of treated kenaf fiber and 1 wt.% EJO plasticizer from 315.53 °C to 290.98 °C for onset temperature and 377.83 °C to 371.17 °C for final degradation temperature. However, after the addition of 5 wt.% EJO to PLA/TK, the thermal stability of PLA was later improved by 4.34 °C. This could have been due to the addition of higher EJO plasticizer content onto the PLA/TK biocomposites. Moreover, this is most likely attributed to the greater dispersion and interaction between polymer and plasticizer, hence, leading to the chain mobility at the higher concentration of EJO in the PLA/TK/EJO biocomposite. 

With alkali treatment, cellulose I from kenaf is converted to cellulose II, which is more thermodynamically stable. Cellulose II has better thermal stability than cellulose I which is likely due to the stronger hydrogen bonds in cellulose II crystal structure as well as the higher purity of the cellulose II. The alkaline solution disrupts the hydrogen bond or hydrophobic interaction in cellulose crystals during the mercerization process and it is an effective process for improving the assembly of cellulose II. This process is beneficial for enhancing the mechanical structure of mercerized cellulose material which is composed of the crystalline structure of cellulose II [94].

Meanwhile, several researchers have reported a decrease in polymers’ thermal stability resulting from the addition of various plasticizers to the polymer and highlighted that the decrease was from the vaporization of the plasticizers [95,96,97]. In this study, the PLA/TK/EJO 5 biocomposite provided a higher decomposition temperature as the addition of the EJO plasticizer improved the adhesion between the PLA and fiber, and thus prevented the migration and leaching of the plasticizer molecules from the PLA matrix at the same time. Hence, this caused a better adhesion between treated kenaf and PLA and affected the decomposition temperature [98]. In view of the obtained results in this study, an optimum concentration of EJO in PLA was achieved at 5 wt.%, whereby a homogeneous blend with good interactions within the blend occurred. Furthermore, an increase in the thermal stability was due to the good dispersion and the interaction with the PLA matrix [66,80].

### 3.5. Differential Scanning Calorimetry Properties

Figure 8 illustrates the DSC thermograms of PLA, PLA/TK and PLA/TK/EJO biocomposites (with various EJO wt.%) and Table 6 reports the glass transition temperature, *T*_g_, cold crystallization temperature, *T*_c_, melting temperature, Tm, enthalpy of crystallization, Δ*H*_c_ and the degree of crystallinity, *X*_c_, for the samples from the DSC results. *T*_g_ was reported as the onset of the glass transition, while *T*_c_ and *T*_m_ were reported as the peak maxima. The degree of crystallinity was calculated using *X*_c_ = Δ*H*_c_/93.7, with 93.7 Jg^−1^ as the melting enthalpy of a PLA crystal infinite size [99,100].

As can be seen from the DSC thermograms of Figure 8, after the addition of treated kenaf fiber to the PLA, it was observed that the *T*_g_ value of PLA decreased from 68.64 °C to 63.75 °C. The reason behind the decrement was due to the introduction of treated kenaf fiber which acted as a nucleating agent to the PLA. The addition of kenaf fibers acting as a nucleating agent to the polymers had been revealed in several past studies [7,62,101]. Cellulose in kenaf plant fiber acts as a nucleating agent to the PLA polymer crystallization. Nucleating agents can effectively promote the crystallization by providing nucleation sites around which the polymer chains can crystallize [102]. Furthermore, the addition of kenaf fibers could accumulate plasticizers on the surface of the fibers by physical interactions and/or chemical linkages leading to the uniform dispersion of plasticizers and improving the plasticization effect [12].

Similar to the *T*_g_ value, the *T*_c_ value for PLA/treated kenaf composite also dropped upon the addition of treated kenaf fiber to the PLA. The reason behind this is the introduction of kenaf fiber hinders the migration and diffusion of PLA molecular chains to the surface of the growing polymer crystals in the composites [103,104]. Thus, it provides a negative effect on the polymer crystallization which later results in a decrease in *T*_c_. The viscosity of the biocomposite mixture increased with the addition of treated kenaf fiber to the polymer which hindered the migration and diffusion of PLA molecular chains in the biocomposite. Furthermore, PLA/TK composite shows a broader cold crystallization temperature range than that of the pure PLA film and the presence of kenaf could promote the cold crystallization behavior of the PLA matrix. The degree of crystallinity, *X*_c_ of the samples increased from 3.63 for PLA to 17.6 for the PLA/TK composite. Kenaf in this study could promote the polymer crystallization on the filler surface and surface of the inorganic fillers [7]. Surface topography of the natural fiber or filler is the crucial factor on crystallinity development [105]. Therefore, the surface roughness of treated kenaf fiber is expected to initiate the growth of crystals on PLA/TK interphase. Greater fiber-matrix interaction had led to the higher degree of crystallinity due to the increment of effectiveness cross-sectional area of the biocomposites caused by treated kenaf fiber particles [106].

Natural fiber had significantly shifted the peak crystallization temperature of polymers to a higher temperature [107]. The nucleating effect of natural fiber had played an important role and influence over the crystallization process of PLA. A similar study was conducted by Suryanegara et al. [108] on the effect of microfibrillated cellulose-reinforcement in PLA who reported that the microfibril acted as a nucleating agent and improved the crystallization behavior of PLA. Furthermore, another study conducted by Ten et al. [109] confirmed the nucleating effect of cellulose nanowhisker (CNW) that resulted in enhanced crystallinity and brought a positive impact in increasing the mechanical properties of PHBV/CNW nanocomposites. In polymer composites, fillers act as an excellent nucleating agent and dominate in altering the crystallization behavior. However, the incorporation of EJO plasticizer at various concentrations did not result in any new peak or major shift of the existing peaks (*T*_g_, *T*_c_, *T*_m1_ and *T*_m2_). The *T*_g_ value of PLA was reduced to 66.49 °C upon addition of PLA/TK/EJO 1 wt.%. The Tg was gradually decreased with the addition of treated kenaf and an increase of EJO loading, and the lowest *T*_g_ was reported at 62.48 °C for PLA/TK/EJO 5 wt.%. The decrement of the *T*_g_ value upon addition of the treated kenaf fiber and EJO plasticizer in PLA/TK/EJO films reduced the cohesive force of attraction between polymer chain, thus penetrating into the PLA matrix which then affected the segmental mobility of polymers.

### 3.6. Plasticizing Effects

A schematic illustration of the proposed mechanism of the plasticization effect can be seen in Figure 9. It can be seen that a polymer network is exposed to plasticizing at an earlier stage. The EJO plasticizer molecules then diffuse into the PLA polymer network and penetrate between the polymer chains and the plasticizer. A plasticizer physically intercalates between the polymer chains and later increases the free volume and decreases the *T*_g_. As the *T*_g_ of the polymer falls below room temperature, the polymer chains move due to the plasticizer molecules acting as a lubricant between the polymer chains.

In addition, with the increasing concentration of plasticizer in the PLA/TK/EJO components, it reduced the free volume of the PLA phase, which then restricted the growth of the PLA spherulite. Therefore, the fall in crystallinity value of the PLA/TK/EJO films was also observed. Several works reported the decrease in the *T*_g_ value of polymers with the addition of natural fibers and plasticizers [110,111]. From their DSC analyses, the decrease in the glass transition temperature (*T*_g_), upon addition of plasticizer had -changed the mechanical behavior of plastic from fragile to ductile and improved the crystallinity of PLA without compromising its biodegradability and biocompatibility [12,66,105].

Furthermore, the addition of treated kenaf fiber and various concentrations of EJO did not cause major changes in the melting behavior of *T*_m1_ and *T*_m2_ of the semi crystalline PLA. Similar trends were also reported by several researchers [12,66,112]. This shows that the EJO plasticizer was comparable in facilitating PLA to form a more thermally stable α-crystal than δ-crystal without affecting the melting point of PLA [113].

## 4. Conclusions

The thermal properties of the treated biocomposites derived from epoxidized jatropha oil, treated kenaf fiber and poly(lactic acid) matrix have been successfully evaluated. The TGA analysis proved that the treated kenaf fiber with EJO (1 to 5 wt.%) slightly increased the thermal stability by an increment in the value of the initial and final decomposition temperature of the biocomposites. The DSC results depicted that the glass transition temperature(*T*_g_) of kenaf fiber-reinforced PLA/EJO biocomposites, was shifted to a lower temperature than that of PLA and significantly affected the crystallization behavior of biocomposites. The result obtained reveal that the new biocomposites, which consisted of treated kenaf reinforced PLA/EJO with good thermal properties, could be developed from the present research as an alternative for nonbiodegradable petroleum-based polymer composites.

## Figures and Tables

**Figure 1 polymers-12-02604-f001:**
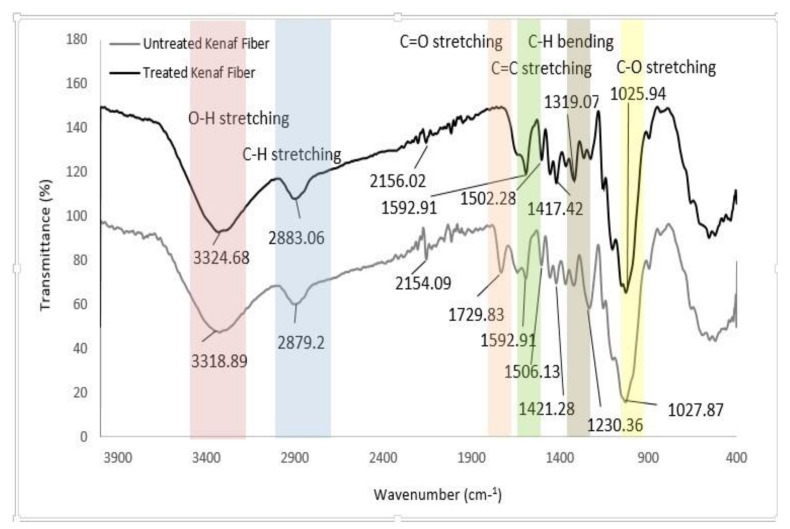
Fourier transform infrared (FTIR) spectra of untreated and treated kenaf fibers.

**Figure 2 polymers-12-02604-f002:**
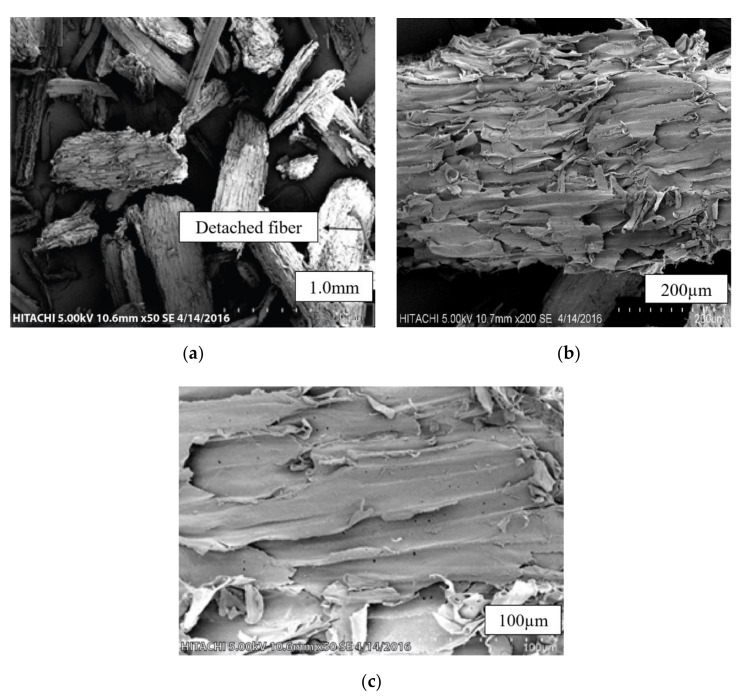
Scanning electron microscopy (SEM) micrographs of untreated kenaf fibers (UTK) at magnifications of (**a**) 50× and (**b**) 200× (**c**) 500×.

**Figure 3 polymers-12-02604-f003:**
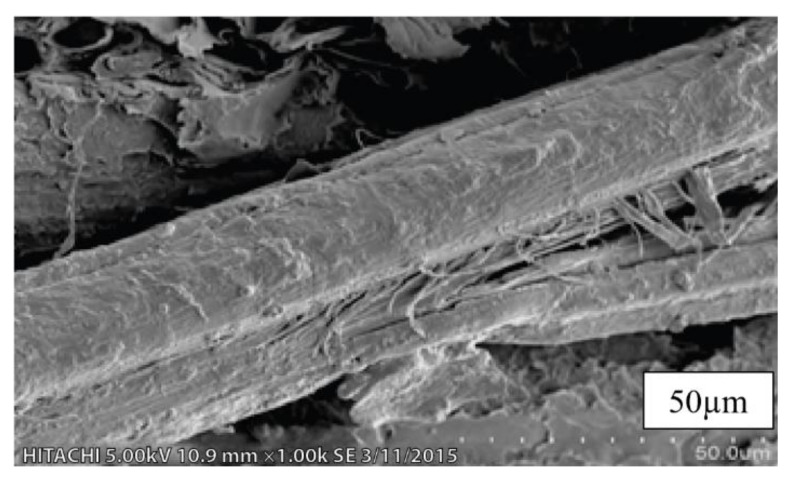
Cross section of untreated kenaf fibers (UTK) at magnification of 1000×.

**Figure 4 polymers-12-02604-f004:**
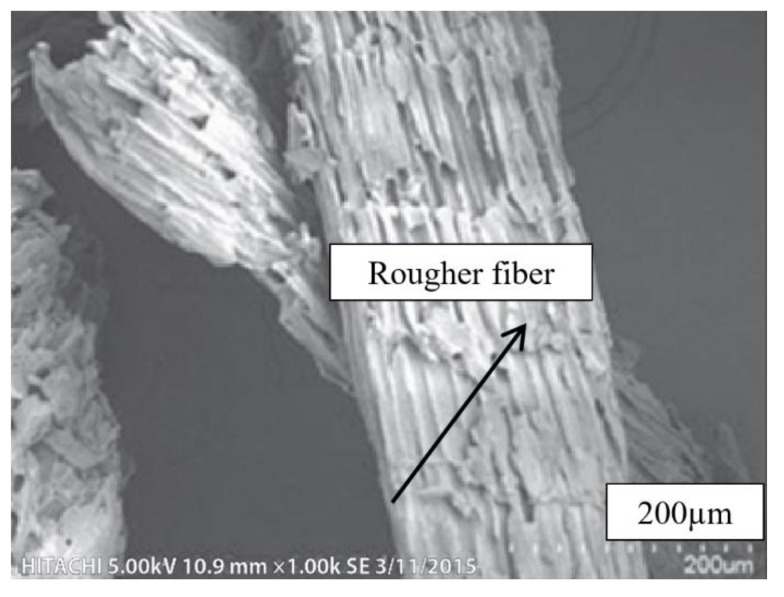
SEM micrograph of treated kenaf fiber (TK) at magnification of 50×.

**Figure 5 polymers-12-02604-f005:**
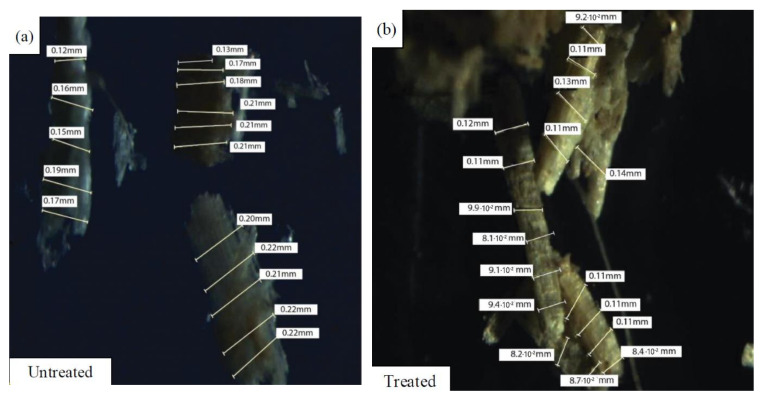
Diameter measurement of (**a**) untreated and (**b**) treated kenaf fibers.

**Figure 6 polymers-12-02604-f006:**
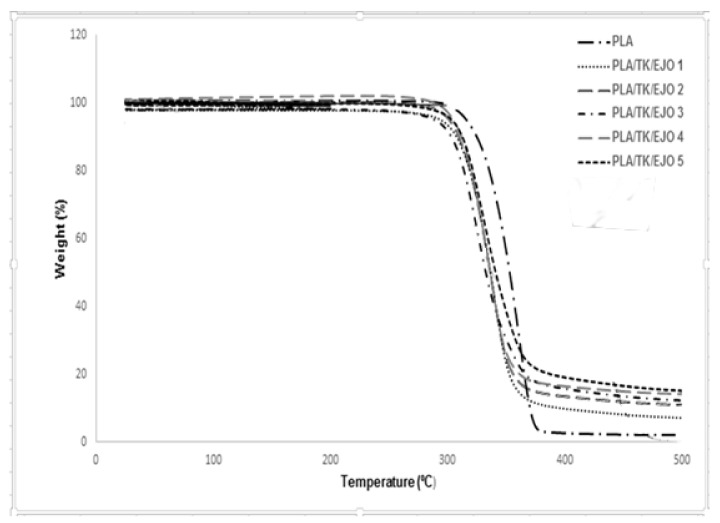
The weight loss (TG) curves of PLA and PLA/TK/EJO biocomposites for various epoxidized jatropha oil (EJO) loadings.

**Figure 7 polymers-12-02604-f007:**
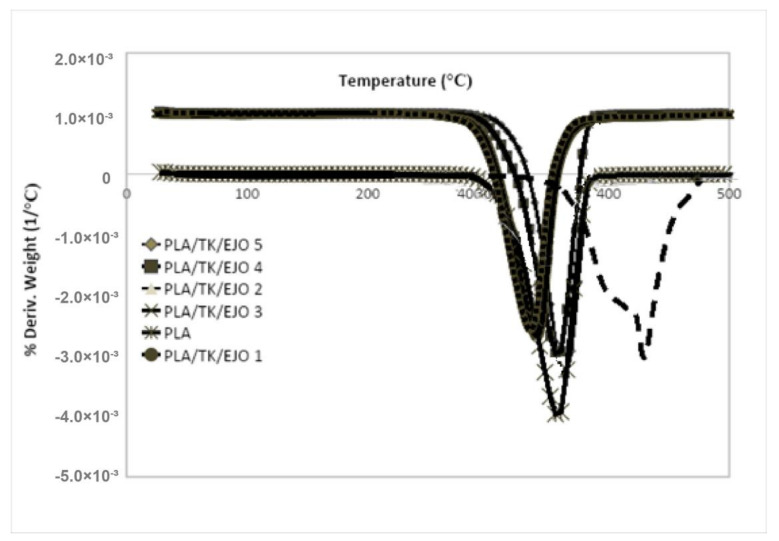
Derivative percentage weight thermogram (DTG) curves of PLA and PLA/TK/EJO biocomposites for various EJO loadings.

**Figure 8 polymers-12-02604-f008:**
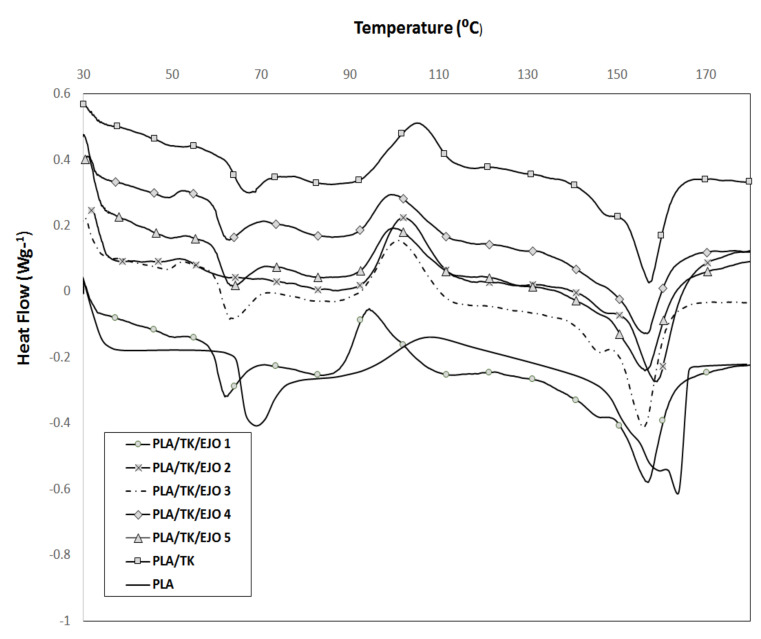
Differential scanning calorimetry (DSC) thermograms of PLA, PLA/TK and PLA/TK/EJO films for various EJO loadings.

**Figure 9 polymers-12-02604-f009:**
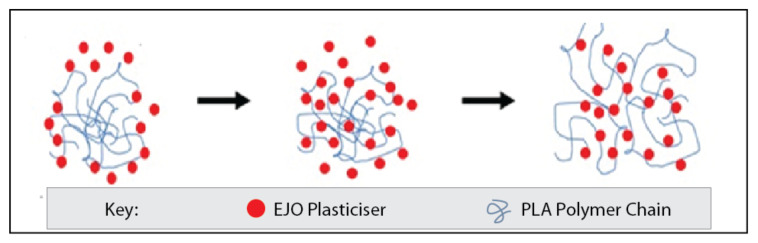
Schematic illustration of the proposed mechanism of plasticization effect.

**Table 1 polymers-12-02604-t001:** Properties of the poly(lactic acid) (PLA) resin from the PLA 2003D datasheet.

Properties	PLA 2003D	American Society for Testing and Materilals (ASTM)
Specific gravity, g/cm^3^	1.24	D792
Notched Izod Impact, J/m	16.0	D256
Melting point, °C	145–160	D3418
Glass transition temperature, °C	55.0–66.0	D3418
Deflection temperature at 0.46 MPa (66 psi), °C	55.0	E2092
D-lactide %	4.0–4.5	

**Table 2 polymers-12-02604-t002:** PLA matrix, kenaf fiber (KF) and epoxidized jatropha oil (EJO) compositions.

PLA Compositions (wt.%)	Kenaf Fiber Compositions (wt.%)	Epoxidized Jatropha Oil Compositions (wt.%)
69	30	1
68	30	2
67	30	3
66	30	4
65	30	5

**Table 3 polymers-12-02604-t003:** Spectra regions for other natural fibers [72].

Bond Type	Hemp	Sisal	Jute	Kapok	This Work
–OH stretching	3448	3447.2	3447.9	3419.7	3318.89, 3324.68
–CH–vibration	2920.5	2942.2	2918.8	2918.1	2879.2, 2883.06
–C=O–stretching		1736.5	1737.2	1741.1	1729.83
–C=C–stretching	1645	1653.9	1653.8	1596.1	1592.91
–CH–bending	1384.1	1384.1, 1259.9	1384.1, 1255.6	1383.6, 1245.5	1319.07, 1230.36
–C–C–stretching	1000–1162	1000–1162	1000–1162	1000–1162	1156–1031
–CH–stretching				897.9	897
–OH	668.9		668.9	668.5	600

**Table 4 polymers-12-02604-t004:** Chemical compositions of untreated (UTK) and treated kenaf fiber (TK).

Chemical Constituents	Compositions (%)	
	Untreated Fiber	Treated Fiber
Cellulose	50.80	55.82
Hemicellulose	20.23	14.91
Lignin	16.75	13.84

**Table 5 polymers-12-02604-t005:** Summary of thermal properties of PLA and PLA/TK/EJO biocomposite samples.

Sample	Onset Degradation Temperature, *T*_onset_ (°C)	Final Degradation Temperature, *T*_final_ (°C)	Rapid Decomposition Temperature, *T*_max_ (°C)	Residue (%)
PLA	315.53	377.83	357.85	2.19
PLA/TK/EJO 1	290.98	371.17	322.15	3.24
PLA/TK/EJO 2	292.20	377.50	324.37	4.79
PLA/TK/EJO 3	293.65	378.17	324.53	4.36
PLA/TK/EJO 4	294.64	378.33	335.13	3.27
PLA/TK/EJO 5	295.99	382.17	336.94	2.91

**Table 6 polymers-12-02604-t006:** Differential scanning calorimetry (DSC) results for PLA, PLA/TK and PLA/TK/EJO films for various EJO loadings.

Sample	*T*_g_ (°C)	*T*_c_ (°C)	*T*_m1_ (°C)	*T*_m2_ (°C)	Δ*H*_c_ (J/g)	*X*_c_ (%)
PLA	68.64	107.75	159.40	163.94	3.40	3.63
PLA/TK/EJO 1	66.49	99.95	147.06	157.16	16.1	17.3
PLA/TK/EJO 2	63.96	99.64	147.67	156.29	15.7	16.9
PLA/TK/EJO 3	63.56	99.12	147.48	155.18	14.8	15.9
PLA/TK/EJO 4	62.96	99.10	147.37	155.83	14.7	15.8
PLA/TK/EJO 5	62.48	99.08	146.02	155.12	11.4	12.2

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
