# Peer review of "Thermal and Structural Analysis of Epoxidized Jatropha Oil and Alkaline Treated Kenaf Fiber Reinforced Poly(Lactic Acid) Biocomposites"

_polymers, 2020, doi:10.3390/polym12112604_

Round 1
Reviewer 1 Report
This paper proposes a new melt-blending processing method for PLA/Kenaf/EJO biocomposites, the content is very detailed. However, there are still some problems to be revised before publication.
1. The citation format of references in the main text, some in '[]' style, while some in '()'.
2. SEM images should be integrated and the scale bar should be as consistent as possible.
3. It is pointed out in the main text that the mechanical properties of biocomposites are improved after alkali treated fibers are added. Here, necessary data to characterize the mechanical properties (such as tensile strength and elongation at break) are necessary.
4. The crystallinity should also be supplemented by XRD.
5. The specific innovation of the article should be highlighted in Introduction.
Author Response
Response to Reviewer 1 Comments
Point 1: The citation format of references in the main text, some in '[]' style, while some in '()'.
Response 1: All the citation format of references in the main text has been changed to ‘[]’ style.
Point 2: SEM images should be integrated and the scale bar should be as consistent as possible.
Response 2: SEM images have been integrated and the scale bar has been standardised.
Point 3: It is pointed out in the main text that the mechanical properties of biocomposites are improved after alkali treated fibers are added. Here, necessary data to characterize the mechanical properties (such as tensile strength and elongation at break) are necessary.
Response 3: The mechanical properties of biocomposites are improved after the alkaline treatment of kenaf fibre. The previous study related to this data has been added in this journal article. (Previous article from our laboratory have reported the mechanical and physical properties of kenaf-reinforced poly (lactic acid) plasticised with epoxidized jatropha oil [3].).
Point 4: The crystallinity should also be supplemented by XRD.
Response 4: While we appreciate the reviewer’s suggestion, our works mainly focus on thermal and structural analysis, the XRD analysis is not the main characterisation for the biocomposite in this study. Additionally, currently our laboratory is also closed and our area is placed under the Conditional Movement Control Order (CMCO) following the nationwide spike in Covid-19 recently. However, we have added additional explanation on the process of crystallinity transition from Cellulose I to Cellulose II from alkaline treatment process in the section of 3.4. Thermogravimetric Analysis Properties.
Point 5: The specific innovation of the article should be highlighted in Introduction.
Response 5: The specific innovation of the article has been highlighted in Introduction. (In this study, the effect of epoxidised jatropha oil and alkaline treated fibre on the thermal properties of PLA biocomposite will be investigated. PLA/Kenaf/EJO biocomposite has been successfully developed as a new environmentally friendly super material with improved properties and cost effective to replace the glass fibre composites. This type of biocomposite complies with the requirements of after-use management of the composites that would not be harmful to the environment. The findings from this study will pave the way towards a greater usage of vegetable oil through epoxidation as well as natural fibre via melt blending for the commercialisation of biocomposites.)
Reviewer 2 Report
The manuscript “Effect of Epoxidised Jatropha Oil and Alkaline Treatment on Thermal Properties of Kenaf Fibre Poly (lactic acid) Epoxidised Jatropha Oil Biocomposite” studies the effect of additon of epoxidised Jatropha oil (EJO) used as plasticizer in a biocomposite. A thermal study was carried out, where the Tg of PLA decrased from 68 to 63 C and the cristallinity increased from 3.6 for PLA to 17.6 for PLA/TK composite. However TGA results are not conclusive.
The manuscript presents, at Introduction, an extensive review of the literature, however it should focus on the topic of interest. At the end of the introduction, it is noted that “there is a lack of published work on using epoxidized jatropha oil (EJO) as a plasticiser for PLA/natural fibre biocomposite”, However, making a quick review it is possible to find studies on EJO and its use as plasticizers in natural fibers (see the list of paper below). There are other observations that would be important to consider
- I think that it is not correct to use Epoxidised Jatropha Oil twice in the title.
- Usually the objective of the study is written at the end of the introduction and not at the abstract.
- The abstract is a brief summary of a reserch articule, It is not necessary to give details of the experimentation for example “Kenaf fiber was treated with 6% of 23 sodium hydroxide (NaOH) solution for 4 hours and the ratio of between matrix, fibre and 24 plasticiser was 69/30/1 wt.%, 68/30/2 wt.%, 67/30/3 wt.%, 66/30/4 wt.% and 65/30/5 wt.% 25 respectively.”.
- It is important to include outstanding results and not expectations in the abstract. For example “it is expected to have greater mechanical locking with matrix” or “further use of non-structural applications.??”
- It is not necessary to reproduce PLA Datasheet (Table 1), use selected properties.
- Briefly describe the epoxidation method in the manuscript, not just refer to another publication
- The reduction of the fiber diameter by NaOH treatment is not clear, Figure 5 does not present a magnification scale line
- The TGA study is very extensive, since it is the main part of the study, however there is no very solid conclusion on the improvement of the thermal stability of the biocomposite prepared.
References:
a). Epoxidized Jatropha Oil as a Sustainable Plasticizer to Poly(lactic Acid)
Polymers 2017, 9, 204; doi:10.3390/polym9060204
b). Epoxidized Vegetable Oils Plasticized Poly(lactic acid) Biocomposites: Mechanical, Thermal and Morphology Properties, Molecules 2014, 19, 16024-16038; doi:10.3390/molecules191016024
c). Producing Jatropha oil-based polyol via epoxidation and ring opening, Industrial Crops and Products, 50, 563-567 (2013)
d). Epoxidation of Jatropha (Jatropha curcas) oil by peroxyacids
Asian Pacific Journal of Chemical Engineering 5(2) 346-354 (2010)
Author Response
Dear Editor
The paper has been fully rewritten and the title changed accordingly to Referee 2. Figures have been updated in order to answer to the well-posed questions of the reviewers. The general structure of the paper has been changed and additional paragraphs introduced. All typos have been corrected.
Response to Reviewer 2 Comments
The manuscript “Effect of Epoxidised Jatropha Oil and Alkaline Treatment on Thermal Properties of Kenaf Fibre Poly (lactic acid) Epoxidised Jatropha Oil Biocomposite” studies the effect of additon of epoxidised Jatropha oil (EJO) used as plasticizer in a biocomposite. A thermal study was carried out, where the Tg of PLA decrased from 68 to 63 C and the cristallinity increased from 3.6 for PLA to 17.6 for PLA/TK composite. However TGA results are not conclusive.
Response: The TGA results has been summarised in the section of 3.4. Thermogravimetric Analysis Properties. (The thermal stability of PLA decreased with the addition of treated kenaf fibre and 1 wt.% EJO plasticiser from 315.53°C to 290.98°C for onset temperature and 377.83°C to 371.17°C for final degradation temperature. However, after the addition of 5 wt.% EJO to PLA/TK, the thermal stability of PLA was later improved by 4.34°C.)
The manuscript presents, at Introduction, an extensive review of the literature, however it should focus on the topic of interest. At the end of the introduction, it is noted that “there is a lack of published work on using epoxidized jatropha oil (EJO) as a plasticiser for PLA/natural fibre biocomposite”, However, making a quick review it is possible to find studies on EJO and its use as plasticizers in natural fibers (see the list of paper below).
Response: While we appreciate the reviewer’s quick review, the listed published works were basically focused on the use of EJO only as a plasticiser for polymers without addition of natural fibres. This study however is basically using epoxidised jatropha oil as plasticiser for PLA, together with the addition of treated kenaf fibre to the PLA composite as well. This research has been studied and developed successfully as new biodegradable biocomposite.
There are other observations that would be important to consider
- I think that it is not correct to use Epoxidised Jatropha Oil twice in the title.
Response: The title has been revised from “Effect of Epoxidised Jatropha Oil and Alkaline Treatment on Thermal Properties of Kenaf Fibre Poly (lactic acid) Epoxidised Jatropha Oil Biocomposite” to “Thermal and Structural Analysis of Epoxidised Jatropha Oil and Alkaline Treated Kenaf Fibre Reinforced Poly (lactic acid) Biocomposite”.
- Usually the objective of the study is written at the end of the introduction and not at the abstract.
Response: The words “objective” of the study has been changed to “main objective” of the study in the abstract section. The later description about the objective of the study has been written at the end of the Introduction section. (In this study, the effect of epoxidised jatropha oil and alkaline treated fibre on the thermal properties of PLA biocomposite will be investigated.)
- The abstract is a brief summary of a reserch articule, It is not necessary to give details of the experimentation for example “Kenaf fiber was treated with 6% of 23 sodium hydroxide (NaOH) solution for 4 hours and the ratio of between matrix, fibre and 24 plasticiser was 69/30/1 wt.%, 68/30/2 wt.%, 67/30/3 wt.%, 66/30/4 wt.% and 65/30/5 wt.% 25 respectively.”.
Response: The abstract has been rewritten and the methodology of the experiment has been simplified and summarised. (Kenaf fibre was treated with 6% of sodium hydroxide (NaOH) solution for 4 hours.)
- It is important to include outstanding results and not expectations in the abstract. For example “it is expected to have greater mechanical locking with matrix” or “further use of non-structural applications.??”
Response: We have revised the abstract to include only outstanding results, not expectations.
- It is not necessary to reproduce PLA Datasheet (Table 1), use selected properties.
Response: Only selected properties which are related with the study in this paper has been rewritten in the PLA properties in Table 1.
- Briefly describe the epoxidation method in the manuscript, not just refer to another publication
Response: The epoxidation method has been described in the manuscript. (A pre‑weighed amount of formic acid was slowly added into a 1L four neck flask where the desired jatropha oil had been prepared and the solution mixture was then heated to 40 °C under continuous stirring in a water bath. Hydrogen peroxide 30 % was then added dropwise to the solution before the reaction temperature was raised up to 60 °C. The epoxidation process was carried out in a closed fume hood.)
- The reduction of the fiber diameter by NaOH treatment is not clear, Figure 5 does not present a magnification scale line.
Response: The Figure of the reduction of fibre diameter by NaOH treatment and the magnification scale line have been clearly presented and updated.
- The TGA study is very extensive, since it is the main part of the study, however there is no very solid conclusion on the improvement of the thermal stability of the biocomposite prepared.
Response: The TGA study has been summarised in the section of 3.4 Thermogravimetric Analysis Properties. (The thermal stability of PLA decreased with the addition of treated kenaf fibre and 1 wt.% EJO plasticiser from 315.53°C to 290.98°C for onset temperature and 377.83°C to 371.17°C for final degradation temperature. However, after the addition of 5 wt.% EJO to PLA/TK, the thermal stability of PLA was later improved by 4.34°C.)
Round 2
Reviewer 1 Report
I recommend the manuscript for publication in this Journal.
Reviewer 2 Report
The manuscript was satisfactorily corrected.
This manuscript is a resubmission of an earlier submission. The following is a list of the peer review reports and author responses from that submission.